# Circadian Regulation of Alternative Splicing of Drought-Associated CIPK Genes in *Dendrobium catenatum* (Orchidaceae)

**DOI:** 10.3390/ijms20030688

**Published:** 2019-02-05

**Authors:** Xiao Wan, Long-Hai Zou, Bao-Qiang Zheng, Yan Wang

**Affiliations:** Research Institute of Forestry; State Key Laboratory of Tree Genetics and Breeding, Chinese Academy of Forestry, Beijing 100091, China; wanxiaoww@163.com (X.W.); zoulonghai@caf.ac.cn (L.-H.Z.); zhengbaoqiang@aliyun.com (B.-Q.Z.)

**Keywords:** alternative splicing, circadian rhythm, CIPK, *Dendrobium catenatum*, drought stress

## Abstract

*Dendrobium catenatum*, an epiphytic and lithophytic species, suffers frequently from perennial shortage of water in the wild. The molecular mechanisms of this orchid’s tolerance to abiotic stress, especially drought, remain largely unknown. It is well-known that CBL-interacting protein kinase (CIPKs) proteins play important roles in plant developmental processes, signal transduction, and responses to abiotic stress. To study the CIPKs’ functions for *D. catenatum*, we first identified 24 CIPK genes from it. We divided them into three subgroups, with varying intron numbers and protein motifs, based on phylogeny analysis. Expression patterns of CIPK family genes in different tissues and in response to either drought or cold stresses suggested *DcaCIPK11* may be associated with signal transduction and energy metabolism. *DcaCIPK9*, -*14*, and -*16* are predicted to play critical roles during drought treatment specifically. Furthermore, transcript expression abundances of *DcaCIPK16* showed polar opposites during day and night. Whether under drought treatment or not, *DcaCIPK16* tended to emphatically express transcript1 during the day and transcript3 at night. This implied that expression of the transcripts might be regulated by circadian rhythm. qRT-PCR analysis also indicated that *DcaCIPK3*, -*8*, and -*20* were strongly influenced by circadian rhythmicity. In contrast with previous studies, for the first time to our knowledge, our study revealed that the major CIPK gene transcript expressed was not always the same and was affected by the biological clock, providing a different perspective on alternative splicing preference.

## 1. Introduction

Drought stress—a major environmental factor—affects the geographical distribution of plants, restricts crop productivity in agriculture, and causes food security problems [1]. *Dendrobium catenatum* is an epiphyte that sticks to the surface of other plants or to naked rocks in the natural environment [2,3]. Because of minimal water interception in these habitats, its roots seriously lack available water and nutrients [4]. To survive the tough environment, *D. catenatum* has evolved succulent pseudobulbs and crassulacean acid metabolism (CAM), a photosynthetic metabolic pathway with high water use efficiency, in response to such a harsh natural environment [2,5,6]. As a highly drought-tolerant species, *D. catenatum* can serve as a model to study survival strategies of plants coping with drought. Studying the gene spectrum of a highly resistant plant that spans evolutionary timescales is the most direct and effective way to implement stress tolerance gene engineering.

The homologous proteins SnRK in green plants, SNF1 in yeast, and AMPK in mammals belong to the SNF1 protein kinase superfamily. Stress-signaling pathways related to yeast SNF1 and mammalian AMPK indicate that the green plant SnRK evolved from energy sensing [7,8,9,10]. The SnRK-CIPK subfamily, the members of which have a self-inhibitory NAF domain in the C-terminus and a serine/threonine protein kinase domain in the N-terminus, is crucial to the calcium signaling pathway. The large number of identified CBL–CIPK combinations suggests that the module is important for transmission of various abiotic stress signals with calcium as a second messenger [11,12]. For example, under low potassium stress, a cytosolic calcium signal activates *At*CIPK23 via *At*CBL1 and *At*CBL9 to phosphorylate and activate the potassium channel AKT1 [13]. Abscisic acid (ABA) accumulation, which is a general response to drought and salt stress, activates the *At*CBL1/9-*At*CIPK26 complex to phosphorylate effector proteins such as RbohF [14]. By overexpression of *OsCIPK23* and cotton *CIPK6*, drought tolerance in Arabidopsis and rice can be enhanced, respectively [15,16].

Given its important roles, the CIPK gene family has been widely characterized in many species, including Arabidopsis [17,18], poplar [18], maize [19], rice [20], soybean [21], and canola [22]. *D. catenatum*, which has a CAM photosynthetic pathway, exhibits high resistance to drought. Yet, its drought resistance strategy is still not clear. To date, a genome-wide analysis of the CIPK gene family in *D. catenatum* has not been performed. The publication of the draft genome sequence of *D. catenatum* enables an analysis of the phylogeny, evolution, structure, and expression of the CIPK gene family [2]. In this study, we identified 24 *Dca*CIPK family members and divided these into intron-poor and intron-rich clades. The gene expression patterns of CIPK genes in different tissues and under drought and cold stress were analyzed using public transcriptome data. Moreover, we examined express patterns for transcripts of CIPK members under drought stress during day and night.

## 2. Results

### 2.1. Genome-Wide Identification of CIPK Gene Family Members in D. catenatum

In total, 24 genes containing both a NAF domain and a kinase domain were identified as *CIPK* gene family members in the orchid. Twenty-two members inferred from the orchid genome annotation [2] and two new CIPK genes (Novl001335 and Novl001612) were newly annotated from novel genetic loci with transcript assembly. The number of amino acid residues of *Dca*CIPK primary proteins ranged from 401 to 502. The relative molecular weights of these CIPK kinase proteins varied from 45.15 to 57.14 KDa. All the proteins had high isoelectric points (pI > 7.0). Detailed information for these genes is listed in Table 1.

### 2.2. Phylogenetic Analysis and Structure of the D. catenatum CIPK Gene Family

Phylogenetic analysis classified the CIPK family members into three subgroups (I, II, and III), whose genes contain nine, five, and ten members, respectively (Figure 1A). To further study the features of the genes (from primary genome annotation), the composition and position of exons, introns, and conserved elements were analyzed. The *Dca*CIPK gene members were clearly divided into an intron-rich clade (>8 intron per gene) and an intron-poor clade [19,23]. All the intron-rich clade members belonged to subgroup I, and those intron-poor clade members related to subgroups II and III (Figure 1B). In subgroups III and II, their genes have introns ranged from zero to four (Figure 1C). In subgroup I, most members contained 13 introns, two members (*DcaCIPK1*, *DcaCIPK21*) contained 12 introns, and only one gene (*DcaCIPK17*) had 11 introns (Figure 1C).

The MEME suite was used to discover conserved motifs of *Dca*CIPK proteins. Thirty motifs were identified using previously determined parameter settings (see Methods). Twelve of those were statistically significant with an E-value less than 0.05. All CIPK proteins contained motifs 1 and 4. Motifs 11, 18, 22, and 26 were only found in members of the intron-rich clade. Motifs 14, 15, 21, 25, 27, 28, and 29 only existed in subgroup II; motifs 12, 13, 16, 17, 19, 20, 23, and 24 were only found in subgroup III; and motifs 11, 18, 22, and 26 were only in subgroup I (Figure 2).

### 2.3. Phylogenetic Analysis of CIPK in Plants

An NJ (neighbor-joining) phylogenetic tree was built to scrutinize the evolutionary relationships of CIPK family members. Eighty-three protein sequences, including those from Arabidopsis, rice, and *D. catenatum*, were divided into five groups (Groups I–V). Proteins from all nine intron-rich *DcaCIPKs* were clustered in Group IV. Including nine *AtCIPKs* and 11 *OsCIPKs*, Group IV was the largest group. Group I comprised seven *DcaCIPKs*, five *AtCIPKs*, and 13 *OsCIPKs*. Group II consisted of three *DcaCIPKs*, four *AtCIPKs*, and four *OsCIPKs*. Group III was the smallest group, containing one *DcaCIPK*, three *AtCIPKs*, and one *OsCIPK*. Finally, four *DcaCIPKs*, five *AtCIPKs*, and four *OsCIPKs* were within Group V (Figure 3).

### 2.4. Expression of D. catenatum CIPK Genes in Different Plant Tissues

To determine the roles of CIPK genes in *D. catenatum*, reassembled transcriptome data was used to illustrate the expression of *DcaCIPKs* in different tissues (root tip, root, flower bud, stem, and leaf; Figure 4; Appendix A). Two primary clusters for expression patterns of *Dca*CIPK genes were found and both of them contain intron-poor and intron-rich genes (Figure 4). Most *Dca*CIPK genes, except *DcaCIPK13*, were detected in these five tissues. Among these detected genes, *DcaCIPK6*, -*15*, and -*18* were the most highly expressed genes in root tip and root; *DcaCIPK1*, -*14*, and -*15* were highly expressed in stem; *DcaCIPK3*, -*14*, and -*19* were highly expressed in leaf; and *DcaCIPK6* and -*18* were highly expressed in flower bud. These results suggested these specific highly expressed genes may have important roles in development or sensing of external signals in the corresponding tissues.

### 2.5. Expression of D. catenatum CIPK Genes under Abiotic Stress

To figure out how *CIPK* gene expression changes in response to abiotic stress, we analyzed the expression patterns of *CIPK* members. Dataset A (see section Materials and methods) represents *DcaCIPK* expression profiles in *D. catenatum* leaves during the day (9:00 am) and night (9:00 pm) under drought treatment (Figure 5A). Dataset B represents *DcaCIPK* expression profiles obtained under cold- and control- treatments (Figure 5B). *CIPK20* and -*21* were significantly fluctuant under drought and cold treatment as compared with normal conditions, respectively.

Under drought stress, *DcaCIPK3*, -*6*, -*12*, and -*16* were upregulated while *DcaCIPK19* and -*20* were downregulated during the daytime (fold change > 1.5). During nighttime, *DcaCIPK3*, -*12*, -*14*, -*16*, and -*20* were induced significantly and *DcaCIPK7*, -*11*, -*18*, -*23*, and -*24* were inhibited (fold change > 1.5) (Figure 5A; Appendix A). During cold treatment, the expression levels of *DcaCIPK1*, -*14*, -*15*, -*18*, and -*19* were increased, while those of *DcaCIPK3*, -*7*, -*11*, -*16*, -*17*, and -*24* were decreased (fold change > 1.5) (Figure 5B; Appendix A). *DcaCIPK3*, -*12*, and -*16* were abundantly expressed during both day and night under drought treatment. *DcaCIPK14* was the only gene that was significantly induced by the two different stress treatments. Ten of the significantly fluctuating expressed genes (*DcaCIPK6*, -*7*, -*12*, -*14*, -*15*, -*16*, -*18*, -*19*, -*20*, and -*23*) were classified into the intron-poor cluster; the rest of the genes, including *DcaCIPK1*, -*3*, -*23*, and -*24*, belonged to the intron-rich cluster.

### 2.6. Alternative Splicing Analysis of CIPK Members under Drought Stress

To study DcaCIPK genes’ expression at transcript levels, we used StringTie to assemble transcripts with genome guide and totally 76 transcripts were reconstructed for the 24 *Dca*CIPK genes (File S1). As shown in the result, 17 *Dca*CIPK genes have multiple splicing variants (Appendix A). We analyzed alternative splicing by examining the structure and expression levels of CIPK genes. *DcaCIPK1* (transcript1, 4, and 5), *DcaCIPK2* (transcript1 and 3), *DcaCIPK3* (transcript1 and 3), *DcaCIPK4* (transcript1), *DcaCIPK5* (transcript1), *DcaCIPK7* (transcript1), *DcaCIPK12* (transcript1), *DcaCIPK14* (transcript1), *DcaCIPK15* (transcript1 and 5), *DcaCIPK16* (transcript1), and *DcaCIPK18* (transcript2) showed significantly diurnal variation in expression levels whether under a drought or moist environment. *DcaCIPK19* (transcript1) and *DcaCIPK24* (transcript1) were only influenced (downregulated) by drought treatment. *DcaCIPK1* (transcript3), *DcaCIPK3* (transcript2 and 5), *DcaCIPK8* (transcript6), *DcaCIPK15* (transcript2), and *DcaCIPK18* (transcript1) were influenced by both water content of the base material (drought and control) and the harvest time (day and night). *DcaCIPK1* (transcript3) at night, *DcaCIPK8* (transcript6), and *DcaCIPK3* (transcript2 and 5) during both day and night, and *DcaCIPK15* (transcript2) during the day were upregulated under drought stress (Figure 6; Appendix A).

Interestingly, we found transcripts in *DcaCIPK1* and *DcaCIPK3* had specific expression patterns: all transcripts in the former gene showed a high abundance in nights but extremely low levels in days (Figure 6B), and the two main transcripts (tanscript1 and transcript3) in the latter gene had a mutually day-and-night reversed expression (Figure 6B). Hence, we analyzed the protein domain of their main expressed transcripts on Pfam according to the hidden Markov model (HMM) profile (Figure 7). Transcript1, 3, and 4 are *DcaCIPK1*′s major isoforms. Besides including the two basic protein domains—pkinase and NAF—KA1 domain is found in transcript1 and 3. Moreover, KA1 is closely adjacent to NAF domain in transcript1 compared with that in transcript3. As for *DcaCIPK3*, transcript2 and transcript5 are the main isoforms of alternative splicing. However, the pkinase domain in transcript2 is incomplete.

Prediction of *cis*-acting regulatory elements indicated that these genes all had elements that are involved with light responsiveness. A circadian element was found in *DcaCIPK2*, -*3*, -*8*, and -*12*. The MBS element, an MYB binding site involved in drought-inducibility, was found in *DcaCIPK5*, -*12*, -*15*, -*16*, and -*19*. *DcaCIPK15* was found to have a low-temperature responsiveness element. In addition, many of these genes contained plant hormone responsiveness elements such as an ABA-responsive element, a methyl jasmonate inducible element, a salicylic acid-inducible element, an ethylene-responsive element, a gibberellin-responsive element, and an auxin-responsive element (Appendix A).

Under drought stress, the expression level of *DcaCIPK3* was upregulated, but the major functional transcript was not the same between day and night. During daytime, transcript5 was the major transcript detected. In contrast, transcript2 was the major transcript at nighttime (Figure 6B, Figure 9C). Promoter region analysis showed *DcaCIPK3* has two circadian rhythm binding sites located at 54 and 84 bp (marked as green boxes in Figure 8).

### 2.7. Effect of Time-of-Day and Drought Stress on Selected Genes mRNA Expression

We selected six genes (*DcaCIPK1*, -*3*, -*7*, -*12*, -*15*, and -*16*) to conduct qRT-PCR validation of their mRNA expression (Figure 9). Time-of-day affected the expression of four selected genes (*DcaCIPK1*, -*3*, -*7*, and -*15*) under drought and control conditions. The expression of *DcaCIPK1*, -*7*, and -*15* exhibited a strong circadian rhythmicity and peaked during the active phase both under drought and control. *DcaCIPK1* significantly elevated during dark phase, while *DcaCIPK7* and *15* were active in day time. The expression levels under drought were lower than that under control both for *DcaCIPK7* and -*15*. *DcaCIPK3* had no obviously difference in day and night, but the gene expression level under drought was higher than that under control. qRT-PCR assays also validated the predicted expression patterns of *DcaCIPK12* and *DcaCIPK16* under drought treatment, respectively. The statistics result shows differences of expression level between drought and control is significant. The average expression level of the two genes under drought stress is higher than that under control, respectively.

## 3. Discussion

How to sense and conduct external stress signals is a basic biological competence in plants. Osmotic stress is the major signal caused by drought stress [10]. As a major calcium sensor protein, CIPK functioning with CBL could transfer the stress signal downstream. The CIPK family has been analyzed in many C3 and C4 plants [18,19], but comprehensive information about CAM plants is limited. *D. catenatum*, a facultative CAM species, has a high level of drought resistance. Previous researches related to this species have mostly focused on polysaccharide hydrolysis and synthesis [24,25], while genes involved in signal transduction have been rarely studied.

### 3.1. Phylogenetics and Structure of DcaCIPK Genes

The phylogenetic analyses of gene structure and evolution provided a theoretical basis for the functional annotation of *Dca*CIPK genes. The intron-rich *CIPK* members from Arabidopsis, rice, and *Dendrobium catenatum* were all clustered in Group IV, whereas the genes in the other four groups were all intron-poor members (Figure 3) [18,20,26]. The closely related orthologous *CIPKs* among different species suggest that an ancestral series of CIPK genes existed before species divergence. The bipolarity of intron numbers in *D. catenatum* genes was also observed in those of other species such as Arabidopsis, maize, and rice; these cases suggest that intron loss and gain events may have happened before the monocot-eudicot divergence [18,19,21].

Previous reports have demonstrated the critical role of CIPK genes under diverse stress, which drove us to collect and analyze the expression profiles of these genes in different tissues and under various abiotic stresses from public transcriptome data. The analysis of global gene expression patterns indicated that the most drought- and cold-inducible genes belonged to the intron-poor cluster (Figure 5). This finding was consistent with the analysis of the CIPK family in soybean [21]. Previous research has indicated that the intron-poor *CIPK* group evolved much later, very likely derived by discarding introns of intron-rich members [21,27]. Therefore, the loss of introns may be an adaption to environmental stress.

### 3.2. DcaCIPK Gene Expression and Functions in Drought Stress

Our results revealed that *DcaCIPK3*, -*12*, and -*16* can be significantly induced during both the day and the night under drought treatment (Figure 5, Figure 6 and Figure 9B). *DcaCIPK*16 is orthologous to *AtCIPK*16 (Figure 3), which can phosphorylate AKT1 to regulate the absorption of potassium by interacting with CBL1/CBL9 [28]. *DcaCIPK16* was thus predicted to have a similar function to *AtCIPK16* in dealing with the iron stress resulting from drought. *DcaCIPK3*, as an orthologous gene of *AtCIPK3*, may play a critical role in ABA signaling and multiple stress responses as shown by studies on *AtCIPK3* [29]. *AtCIPK3* can be rapidly activated by salt, drought, cold, and osmotic stress treatments. Both drought and salt have a common ABA-dependent pathway. Disruption of *AtCIPK3* altered the expression patterns of stress-responsive genes (*RD29A* and *KIN1/KIN2*) triggered by cold, high salt, and ABA. However, the disruption of *AtCIPK3* did not affect the expression of genes (*RD29A* and *KIN1/KIN2*) induced by drought, suggesting that *AtCIPK3* acts in a distinct way in response to drought stress. On account that cold-induced gene expression has been shown to be independent of ABA production, these results indicated *AtCIPK3* represents a cross-talk node between the ABA-dependent and ABA-independent pathways in stress responses [29,30]. The precise role of *DcaCIPK12* cannot be elucidated from evolutionary relationships because the function of neighboring *AtCIPKs* (*AtCIPK12*/*19*) is still unclear. These genes were upregulated during both the day and the night under water deficit stresses and therefore are forecast to play a critical role in drought resistance.

*DcaCIPK14* was significantly upregulated under two types of abiotic stress, which indicated this gene presumably had a functional overlap with responses to both water deficit and low-temperature stress (Figure 5). Moreover, *DcaCIPK14* was abundantly expressed in pseudobulbs and leaves, which are rich in polysaccharides (Figure 4). Plentiful polysaccharides could supply a large amount of bound water and also hydrolyze into soluble monosaccharides in case of severe abiotic stress [31,32,33]. Previous studies have indicated that the CIPK gene may have a link with the sucrose content of plants. For example, *ScCIPK8*, a sugarcane CIPK gene, had a negative correlation with sucrose content of leaves [34,35]. Light-interruption stress, which inhibits the synthesis of sucrose, reduces the expression of *AtCIPK14* expression; after feeding sucrose, however, the expression of *AtCIPK14* goes up [36]. As an orthologous gene of *AtCIPK14*, *DcaCIPK14* may act in connecting and regulating stress signal induction and energy metabolism.

### 3.3. Circadian Rhythm and Drought Stress Both Influence Alternative Splicing of CIPK Members

The abundance analysis for *Dca*CIPK genes’ alternative splicing isoforms indicated that circadian rhythm influences expression levels of transcripts. *DcaCIPK3* may play a significant role in drought response similar to the orthologous gene *AtCIPK3*. The overall abundance of *DcaCIPK3* transcripts and the time-of-day qRT-PCR results also supported the important role of this gene in drought stress. *AtCIPK3* has the highest number of splice variants of *CIPK* members in *A. thaliana*, and two of the transcripts (*AtCIPK3.1* and *AtCIPK3.4*) were more highly induced under ABA and drought treatment [37]. Similarly, preference of specific transcripts was also found in *D. catenatum*. However, it seemed this phenomenon was more likely raised by circadian rhythm in our experiment. Under drought treatment, the expression levels of different transcripts suggested that the most highly expressed transcript variant of *DcaCIPK3* tended to be transcript1 during the daytime and transcript3 at nighttime. The moist treatment had a negative effect on the sum expression abundance of various transcripts, but did not change the routine preference of transcript1 during the day and transcript3 at night, respectively. The analysis of transcript isoforms and the prediction of *cis*-acting regulatory elements of *DcaCIPK3* indicated that circadian rhythm may influence the preference of alternative splicing (Figure 6). *DcaCIPK3* may use different splicing ways during day and night to cope with drought stress. *DcaCIPK3* represents a critical role for improving tolerance of *D. catenatum* to drought and deserves further analysis to reveal its functional roles in drought response and the underlying molecular mechanisms for the preference of splicing, which is currently underway.

Diurnal variation in gene expression levels might be ascribed to the impact of biological clocks. If the sampling time had been fixed during the day, the association with drought tolerance of genes such as *DcaCIPK1*, -*7*, and -15 (Figure 6 and Figure 9 and Appendix A) would have been missed, because its expression level was nearly negligible. Therefore, the choice of sampling time should be taken into consideration in terms of the influence of the bioclock.

Alternative splicing increases gene functional diversity by regulating alternative exon recognition [38]. Our analysis indicated that this combinatorial control mechanism greatly influenced the protein domain in main expressed transcript (Figure 7). Different domain combinations may undertake different duties. Incomplete pkinase domain in transcript2 of *DcaCIPK3* may cause losing function of phosphorylation which eventually prevents signal downward transduction in night time.

As a facultative CAM plant, the specific photosynthetic pathway may also have a great influence on the induction of *DcaCIPK* splice variants. How *Dca*CIPK genes are influenced by the biological clock and how drought stress affects the abundance of transcripts should be further studied.

## 4. Materials and Methods

### 4.1. Genome-Wide Identification of the CIPK Gene Family in D. catenatum

The *D. catenatum* genome (assembly ASM160598v1) [2] and its annotation (general feature format, GTF) were downloaded from the National Center for Biotechnology Information (NCBI) to extract protein and CDS sequences for all genes. To identify CIPK proteins, the hidden Markov model (HMM) profile of NAF with the signature domains (PF03822) was downloaded from Pfam [39] and used to search by HMMER 3.0 [40]. Each *Dca*CIPK protein sequence was examined for the presence of the serine/threonine protein kinase domain in the N-terminus to be considered as a member of the *D. catenatum* CIPK family. The putative CIPK family members were further reviewed by Pfam and SMART software [41]. Molecular weight (MW) [42] and isoelectric point (pI) [43,44] of CIPK members were calculated by “Compute pI/Mw function”of ExPASy [45].

### 4.2. Multiple Protein Sequence Alignment and Phylogenetic Tree Construction

The protein sequences of all *Dca*CIPK family members were aligned using Muscle [46] with default parameters, and a phylogenetic tree was constructed using neighbor-joining (NJ) [47] method built-in MEGA7.0 [48]. The bootstrap values for phylogenetic trees were based on 1000 replicates. We downloaded amino acid sequences for 26 *At*CIPK [14,17] and 33 *Os*CIPK [20] proteins from NCBI and pooled them with sequences of 24 *Dca*CIPK proteins. An NJ phylogenetic tree for protein sequences from *D. catenatum*, *Arabidopsis thaliana*, and *Oryza sativa* was constructed by using the same procedure described above, and pairwise deletion and the Poisson model were introduced.

### 4.3. Exon–Intron Structure Analysis and Identification of Conserved Motifs

Visualization of gene features was performed using the gene structure display server by mapping the sequences coding for amino acids in proteins (CDS) to genomic sequences [49]. Motifs were discovered by MEME [50]. The parameters were set as follows; number of repetitions, any; maximum number of motifs, 30; optimum motif width, between 6 and 200 residues.

### 4.4. Transcriptome Analysis of D. catenatum CIPK Gene Expression

The expression patterns of CIPK gene family members in *D. catenatum* under different stress treatments (drought [51], Dataset A; low temperature [31], Dataset B) and in different tissues (Dataset C) [52] were analyzed (Table 2). Transcriptome data for drought stress, cold treatment, and specific tissues were downloaded from the Sequence Read Archive (SRA) database at the NCBI.

The raw data was stripped of adapters and low-quality reads (and bases) and then rRNA and virus data were filtered out, which were performed by using software Fastq_clean [53] with the settings described in our previous work [51]. The clean reads were aligned to the *D. catenatum* genome using Hisat2 [54,55] with options: -dta and –no-unal. The aligned outputs were converted from SAM to BAM format by using samtools [56]. The “htseq-count” function of Python package Htseq [57] was used to calculate the read-count of genes with default parameters. Expression differences among gene family members were further analyzed by using the R package DESeq2 with relative log expression (RLE) normalization and an adjusted *p*-value cutoff 0.05 [58]. HemI 1.0 [59] was used to illustrate heatmaps with settings: average linkage for clustering and Pearson distance for similarity metric.

### 4.5. Alternative Splicing Analysis and Cis-Acting Regulatory Element Prediction

As the protocol suggested by Pertea et al. [55], the BAM files described above were used as inputs for StringTie [60] to assemble and quantify transcripts of each gene. The visualization of transcripts, including structure and expression levels, was achieved using R [61] package Ballgown [62]. Prediction of *cis*-acting regulatory elements in promotor sequences was performed by using PlantCARE [63]. The sketch of the domains in main expressed transcripts was obtained at Pfam by inputting transcripts amino acid sequences [39].

### 4.6. Plant Material, Growth Conditions, and Experimental Treatments

*D. catenatum* plants were grown in a 5-cm diameter plot in an artificial climate chamber maintained under 12 h light (light intensity ~100 μmol·m^−2^s^−1^, 28 °C, 60% relative humidity)/12 h dark (22 °C, 70% relative humidity) cycles. Irrigation was withheld from the drought group until the volumetric water content of the base material dropped to 0%. The volumetric water content of base material was kept at between 30% and 35% in control group and ~0% in drought group. The topmost mature leaves, generally the fourth and fifth leaves, were harvested with a two-hour interval in a day (8:00 am–24:00 pm–6:00 am) and immediately frozen in liquid nitrogen. Moreover, we also sample specifically at 9:00 AM at day and 9:00 PM at night. These samples were for Quantitative Real-Time PCR (qRT-PCR) assessments. Each experimental treatment was performed in four replicates.

### 4.7. RNA Extraction

RNA was isolated from ground tissue using the RNAprep Pure Plant Kit (Polysaccharides & Polyphenolics-rich, TIANGEN Co. LTD, Beijing, China) according to the manufacturer’s protocols. RNA integrity was analyzed on a 2100 Bioanalyzer (Agilent Technologies, Santa Clara, CA, USA).

### 4.8. Quantitative Real-Time PCR

Primer pairs (Appendix A) were designed by Primer 3. qRT-PCR experiments were performed using Roche LightCycler^®^ 480 II system (LC480, Basel, Switzerland) with TB Green Premix Ex Taq (Takara Bio. Inc., Kusatsu, Japan). Each experimental treatment was performed in four replicates. The relative expression level was calculated as 2-∆∆*C*t and normalized using 18s as internal standard.

### 4.9. Statistical Analysis 

Data are presented as means ± SEM. To determine the effect of the volumetric water content of the base material (drought and control) and the harvest time (day and night) on the expression level of *DcaCIPK* transcripts, a two-way ANOVA was performed using Statistical Analysis System 9.4 (SAS) and then followed by a Duncan’s test (*p* < 0.05) to determine if there was a statistical difference between the mean expression levels of different isoforms under different treatments [64]. The time-of-day effect was evaluated by ANOVA one-way for drought and control situations followed by LSD post hoc analysis. *p* < 0.05 was considered significant.

## Figures and Tables

**Figure 1 ijms-20-00688-f001:**
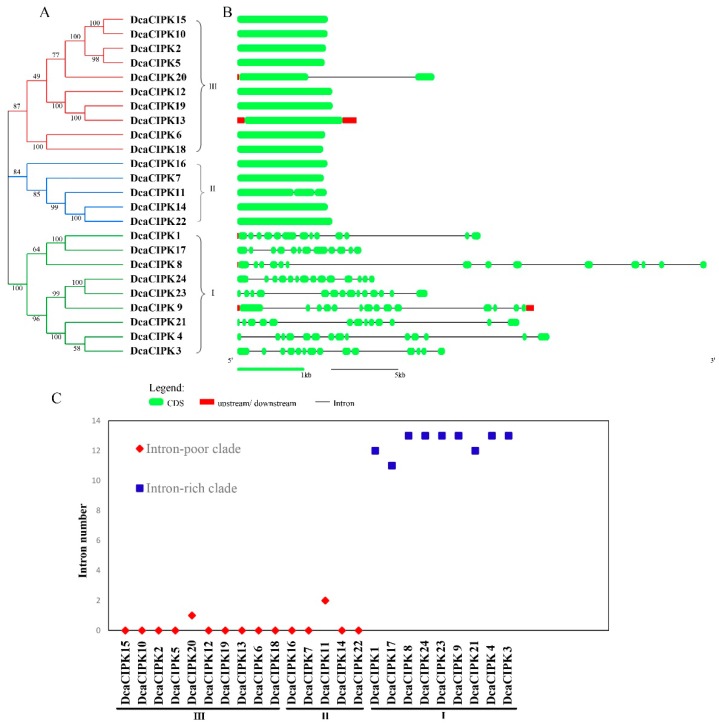
Phylogenetic relationship of *D. catenatum* CIPK proteins and gene structure. (**A**) Neighbor-joining tree. *D. catenatum* CIPK genes were divided into three subgroups (I–III) with different colored branches. (**B**) Exon and intron analysis was performed using GSDS. Green boxes represent exons and black lines represent introns. Red boxes represent upstream/downstream-untranslated regions. (**C**) Classification of CIPK genes into an intron-poor clade (red diamond) and an intron-rich clade (blue squares).

**Figure 2 ijms-20-00688-f002:**
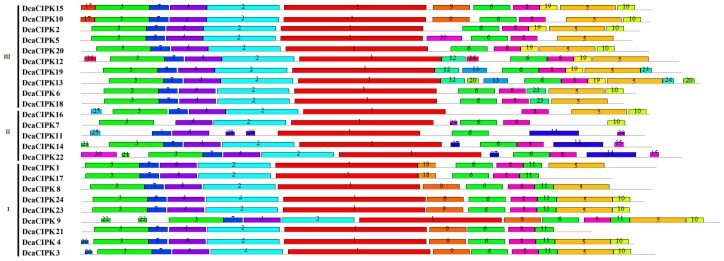
Conserved motifs in *D. catenatum* CIPK proteins. All motifs were identified by MEME with the 24 complete protein sequences of the CIPKs.

**Figure 3 ijms-20-00688-f003:**
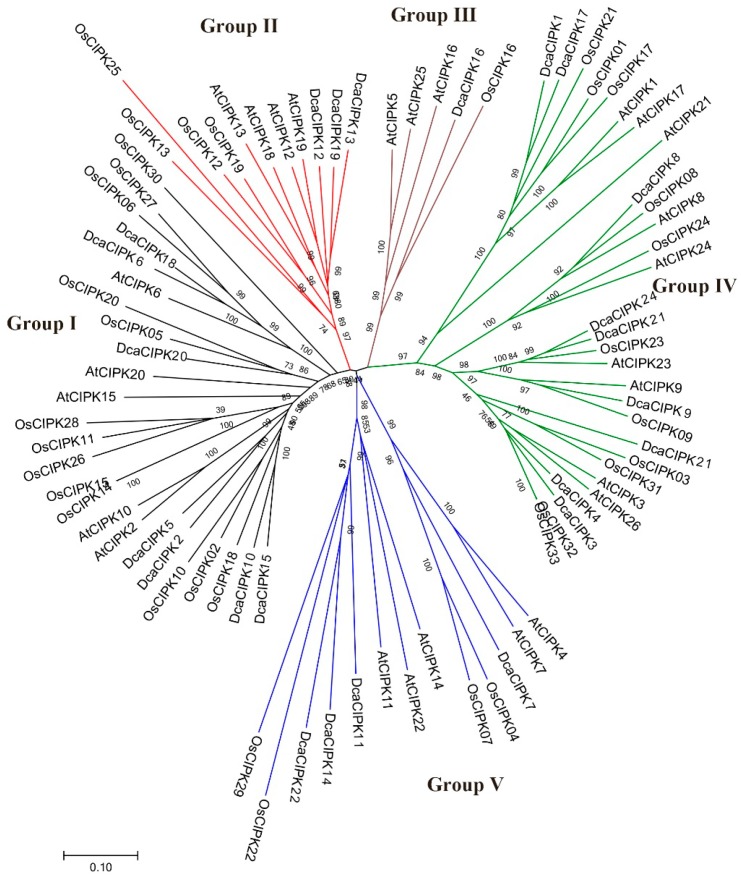
Evolutionary analysis of *Dca*CIPK proteins. Full-length sequences of 83 CIPK proteins from *D. catenatum*, Arabidopsis, and rice were used to construct the phylogenetic tree using MEGA7 with the NJ method. Subfamilies (I–V) are highlighted with different colors.

**Figure 4 ijms-20-00688-f004:**
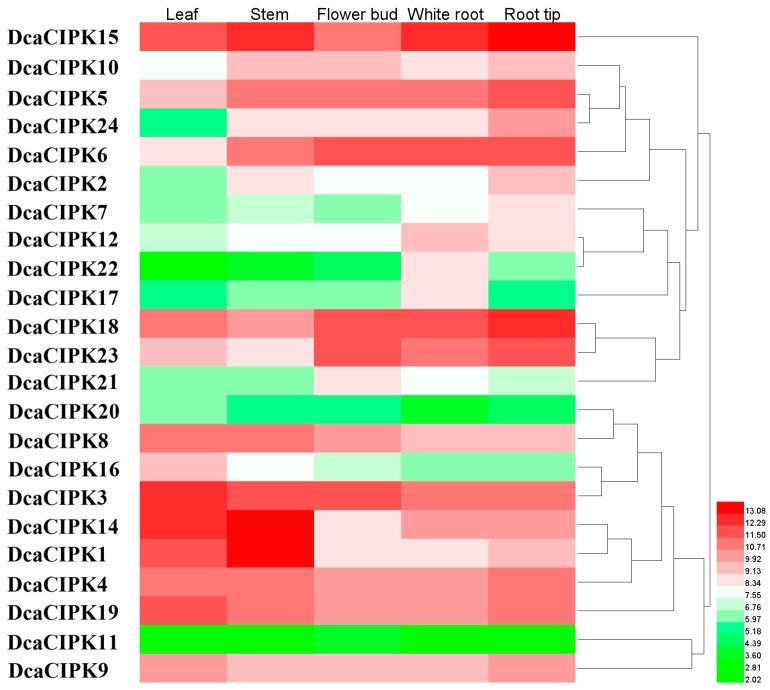
Heatmap of expression profiles for *Dca*CIPK genes in different tissues. Read-count values were normalized with logarithm base two. The scale represents the relative signal intensity of read-count values.

**Figure 5 ijms-20-00688-f005:**
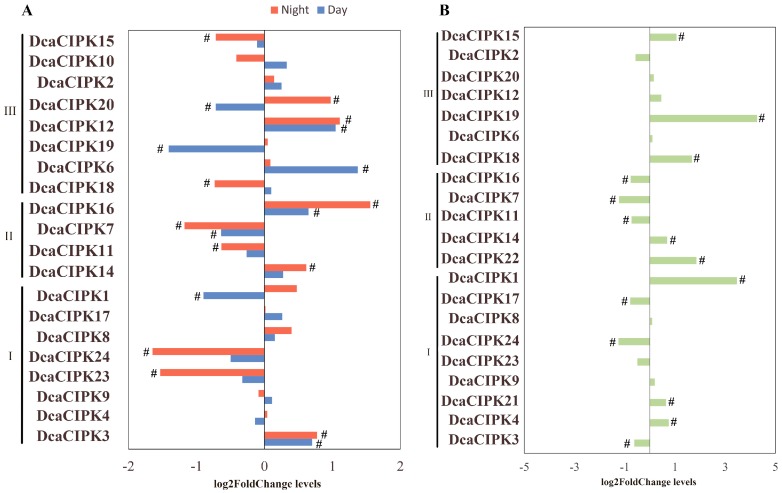
(**A**) Expression levels of *Dca*CIPK genes under drought stress vs (normal condition, moist matrix). (**B**) Expression levels of *Dca*CIPK genes under cold stress (vs. control). Adjusted *p*-value < 0.05. ^#^ indicates fold change >1.5.

**Figure 6 ijms-20-00688-f006:**
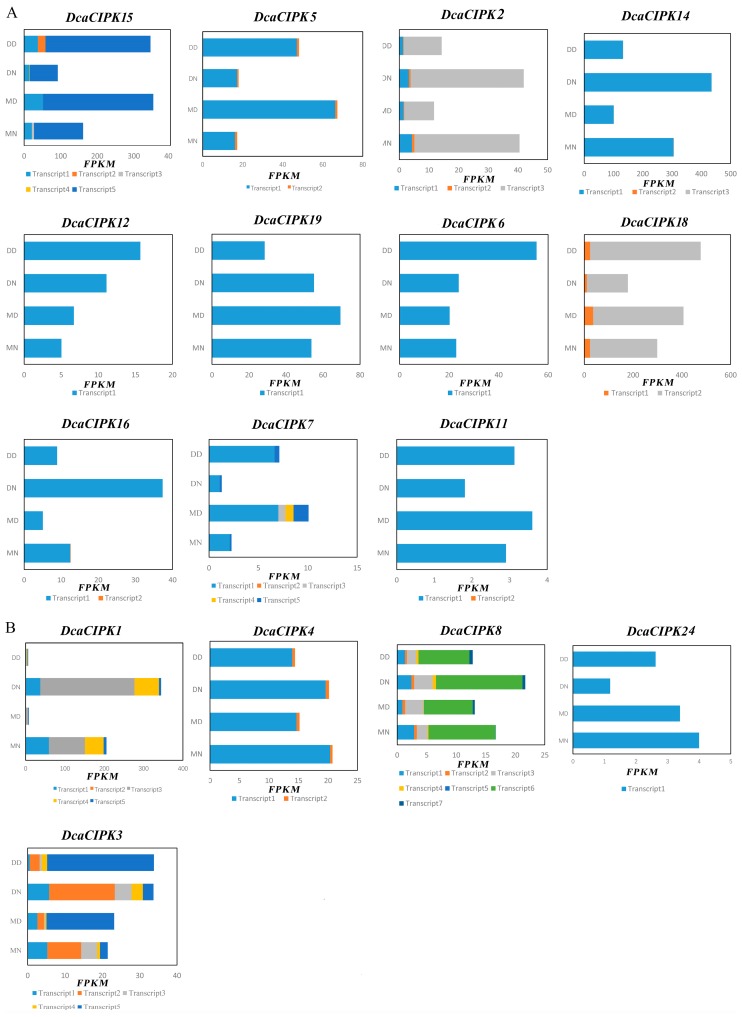
Abundance of *DcaCIPK*s’ alternative splicing isoforms under drought stress. (**A**) Intron-poor group. (**B**) Intron-rich group. DD represents drought treatment in daytime; DN represents drought treatment in nighttime; MD represents moist treatment in daytime; MN represents moist treatment in nighttime.

**Figure 7 ijms-20-00688-f007:**
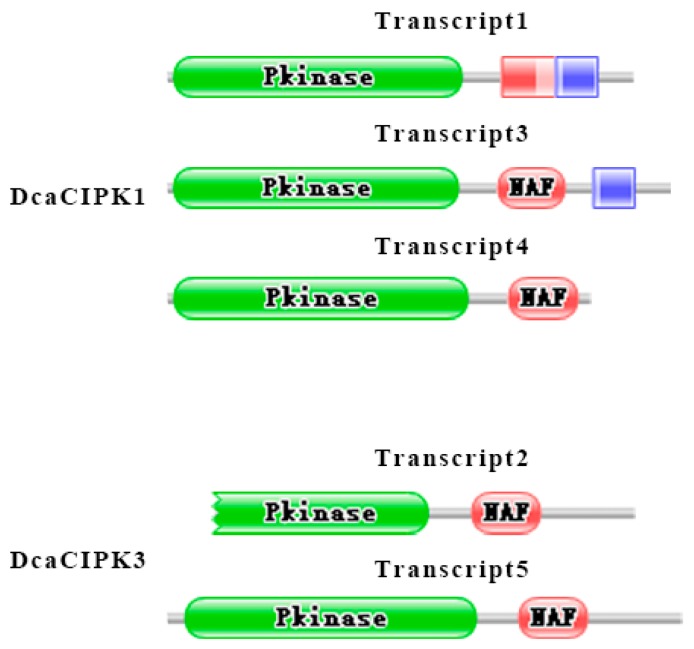
Analysis of the protein domains in main expressed transcripts of *DcaCIPK*1/3. Green boxes represent pkinase domain. Red boxes represent NAF domain. Blue boxes represent KA1 domain. Grey lines represent amino acid sequences.

**Figure 8 ijms-20-00688-f008:**
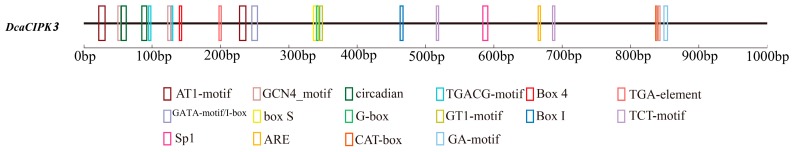
Analysis of the important *cis*-regulatory elements in the sequence 1 kb upstream of *DcaCIPK3*. AT1-motif, part of a light responsive module; GCN4 motif, *cis*-regulatory element involved in endosperm expression; circadian, *cis*-acting regulatory element involved in circadian control; TGACG-motif, *cis*-acting regulatory element involved in MeJA responsiveness; Box 4, part of a conserved DNA module involved in light responsiveness; TGA-element, auxin-responsive element; GATA-motif/I-box, part of a light-responsive element; box S, wounding and pathogen responsiveness; G-box, *cis*-acting regulatory element involved in light responsiveness; GT1-motif, light-responsive element; Box I, light-responsive element; TCT-motif, part of a light-responsive element; Sp1, light-responsive element; ARE, *cis*-acting regulatory element essential for anaerobic induction; CAT-box, *cis*-acting regulatory element related to meristem expression; GA-motif, part of a light-responsive element.

**Figure 9 ijms-20-00688-f009:**
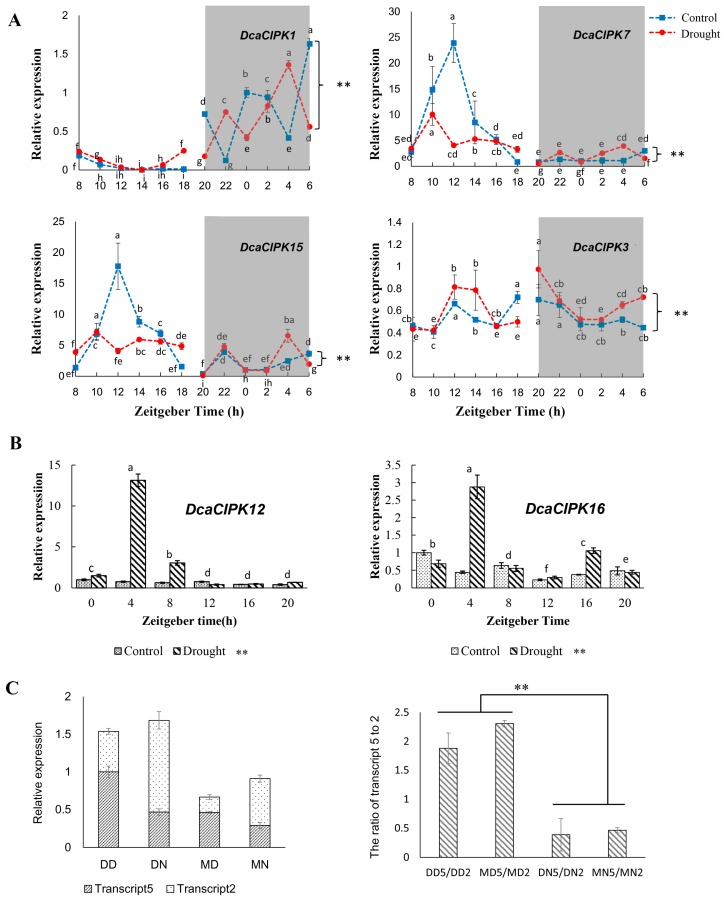
(**A**) Effect of time-of-day on *Dca*CIPK genes expression induced by drought stress. Different letters above the line points indicate statistically significant differences among sampling time treatment at the same base material water content level (*p* < 0.05). (**B**) Effect of drought treatment on *Dca*CIPK gene expression. Light phase began at Zeitgeber Time 6 (ZT6) and dark phase (shaded areas) began at ZT18. Data are presented as mean ± SEM. Experiments were performed 4 times. * indicates differences the effect of base material water content is significant. Different letters above the column indicate statistically significant differences among sampling time treatment (*p* < 0.05). (**C**) Relative abundance of drought treatment on transcripts (2 and 5) of *DcaCIPK3* expression. DD represents drought treatment in daytime; DN represents drought treatment in nighttime; MD represents moist treatment in daytime; MN represents moist treatment in nighttime. “2” and “5” after “DD, MD, MN, and MN” represent transcript2 and 5, respectively.

**Table 1 ijms-20-00688-t001:** List of 24 CIPK genes identified in *D. catenatum* genome and their sequence characteristics.

Name	Gene ID	Accession ID in NCBI	Locus	Gene Length (bp)	Amino Acid Length (aa)	PI	Mw (KDa)	Exons	CDS Length (bp)
*DcaCIPK1*	Dca001684	XP_020700642.1	Dcat_scaffold_83:1213297-1226000	12,704	452	7.94	51.17	13	1359
*DcaCIPK2*	Dca021183	XP_020683888.1	Dcat_scaffold_6544:136242-137564	1323	440	9.16	50.23	1	1323
*DcaCIPK3*	Dca019377	XP_020701813.1	Dcat_scaffold_8095:113831-123888	10,058	451	8.7	51.47	14	1356
*DcaCIPK4*	Dca004523	XP_020703034.1	Dcat_scaffold_4350:650622-668668	18,047	434	8.32	49.36	14	1305
*DcaCIPK5*	Novl001335	XP_020700988.1	Dcat_scaffold_787:157699-159003	1305	432	9.18	49.55	1	1299
*DcaCIPK6*	Dca007455	XP_020673946.1	Dcat_scaffold_3166:141500-142810	1311	436	9.1	48.19	1	1311
*DcaCIPK7*	Dca005524	XP_020676722.1	Dcat_scaffold_1341:459825-461114	1290	429	9.32	48.2	1	1290
*DcaCIPK8*	Dca002402	XP_020676379.1	Dcat_scaffold_2414:350624-380208	29,585	449	8.01	51.09	14	1350
*DcaCIPK9*	Dca000041	XP_020685625.1	Dcat_scaffold_358:1253684-1269205	15,522	502	9.24	57.14	14	1509
*DcaCIPK10*	Dca024760	XP_020681762.1	Dcat_scaffold_45917:29214-30560	1347	448	9.07	50.97	1	1347
*DcaCIPK11*	Dca000484	XP_020701016.1	Dcat_scaffold_787:369843-371213	1371	443	8.71	49.58	3	1332
*DcaCIPK12*	Dca017399	XP_020676379.1	Dcat_scaffold_2131:315817-317232	1416	471	8.79	53.46	1	1416
*DcaCIPK13*	Novl001612	XP_020695665.1	Dcat_scaffold_937523:77342-78802	1847	486	8.01	54.74	1	1461
*DcaCIPK14*	Dca010961	XP_020686786.1	Dcat_scaffold_19096:431277-432626	1350	449	8.28	50.5	1	1350
*DcaCIPK15*	Dca024725	XP_020705240.1	Dcat_scaffold_18826:50805-52154	1350	449	9.25	50.96	1	1350
*DcaCIPK16*	Dca007453	XP_020673947.1	Dcat_scaffold_3166:100617-101960	1344	447	8.59	49.78	1	1344
*DcaCIPK17*	Dca018676	XP_020695665.1	Dcat_scaffold_937621:2927-7179	4253	417	7.57	47.27	12	1250
*DcaCIPK18*	Dca003241	XP_020675333.1	Dcat_scaffold_9707:650080-651360	1281	426	9.04	47.15	1	1281
*DcaCIPK19*	Dca021168	XP_020702232.1	Dcat_scaffold_1902:144820-146241	1422	473	7.99	53.5	1	1422
*DcaCIPK20*	Dca003305	XP_020702013.1	Dcat_scaffold_3734:1066698-1076027	9330	446	9.37	49.92	2	1341
*DcaCIPK21*	Dca016061	XP_020688235.1	Dcat_scaffold_114:141330-157531	16,202	401	8.78	45.15	13	1206
*DcaCIPK22*	Dca021931	XP_020680456.1	Dcat_scaffold_8676:215192-216610	1419	472	9.3	52.82	1	1419
*DcaCIPK23*	Dca012740	XP_020697021.1	Dcat_scaffold_2959:424009-432987	8979	432	8.94	48.82	14	1299
*DcaCIPK24*	Dca025565	XP_020688792.1	Dcat_scaffold_11189:59465-64362	4898	442	8.78	49.93	14	1329

**Table 2 ijms-20-00688-t002:** Treatments and accessions of biosamples in this study.

Dataset	Tissue	Treatment	Collected Time	BioSample Accessions in NCBI	Sources
Dataset A	Leaf	Drought stress	Day (9:00 AM)	SAMN08512102–SAMN08512105	Wan et al. [51]
			Night (9:00 PM)	SAMN08512110–SAMN08512113	
		Control	Day	SAMN08512106–SAMN08512109	
			Night	SAMN08512114–SAMN08512117	
Dataset B	Leaf	Cold stress	Day	SAMN04534730–SAMN04534732	Wu et al. [31]
		Control	Day	SAMN04534727–SAMN04534729	
Dataset C	Flower buds	n.a.	Day	SAMN05908201	Zhang et al. [52]
	Leaf	n.a.	Day	SAMN05912851	
	Green root tip	n.a.	Day	SAMN05908239	
	White root	n.a.	Day	SAMN05908241	
	Stem	n.a.	Day	SAMN05908200	

“n.a.” indicates that the current item is not available.

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
