# Peer review of "Circadian Regulation of Alternative Splicing of Drought-Associated CIPK Genes in Dendrobium catenatum (Orchidaceae)"

_ijms, 2019, doi:10.3390/ijms20030688_

Reviewer 1 Report

The authors have addressed most of my concerns and the manuscript looks pretty much improved. Only a minor suggestion is that they should spend more time and efforts in editing the text. For instance, Line 17: "DcaCIPK11 may associated with"- I think it should be "DcaCIPK11 may be associated with". Such minor language problems are still present in the current version. After the another round of language editing, the manuscript should be acceptable. 

Author Response

Dear Madam/Sir:

Thank you for your valuable comments regarding our manuscript to International Journal of Molecular Sciences (Paper ijms-439931). Our responses to your comments and suggestions are the following:

Comment 1: The authors have addressed most of my concerns and the manuscript looks pretty much improved. Only a minor suggestion is that they should spend more time and efforts in editing the text. For instance, Line 17: "DcaCIPK11 may associated with"- I think it should be "DcaCIPK11 may be associated with". Such minor language problems are still present in the current version. After the another round of language editing, the manuscript should be acceptable.

Response: We have corrected the mistakes in Line 17 following your suggestion. In addition, we have finished another round of language editing to our manuscript.

Reviewer 2 Report

General comments:

In general, the changes implemented by the authors deeply improve and clarify the manuscript compared to the first version I reviewed. The experimental approach sounds well designed and appropriate. The results are well described, well analyzed, and well commented in the discussion section.

However, some minor and very minor comments are listed below.

Minor but important comment:

-          While the description for assembly of transcript isoforms was improved in the present version of the manuscript, the raw results of this analysis is still missing. I do suggest to summarize the results in table (or a supplementary table) that would show the number of splicing variants for each DcaCIPK genes, the number of exons, the length of each transcript isoform and any other key features. Moreover, these important results have to be described and commented

Very minor comments:

-          Figure 4 : a hierarchical clustering was performed together with the heatmap visualization of the DcaCIPK genes expression profiles. The HClust parameters have to be mentioned (distance, linkage method). Moreover, the resulting clustering is not commented.

-          Page 4, line 82 : “[…] their genes have introns raged from zero to four.” I guess the authors meant “ranged”

-          Page 7, line 142: the authors used the words “volatile genes”. I think that, in this context, this adjective can’t be used for qualifying “genes”

-          Pages 11-12, lines 217-218 : “qRT-PCR assays also validated the predicted function of DcaCIPK12 and DcaCIPK16 under drought treatment”. While qRT-PCR can’t validate a protein/gene function, please rephrase without using “function”

-          Page 14, line 320 : start a new line for “4. Materials and methods”

-          Page 16, line 362 : “Heml 1.0”, according to the reference No 59, the spelling is HemI (uppercase “i” not lowercase “l”)

Author Response

Dear Madam/Sir:

Thank you for your valuable comments regarding our manuscript to International Journal of Molecular Sciences (Paper ijms- 439931). Our responses to your comments and suggestions are the following:

Comment 1: While the description for assembly of transcript isoforms was improved in the present version of the manuscript, the raw results of this analysis is still missing. I do suggest to summarize the results in table (or a supplementary table) that would show the number of splicing variants for each DcaCIPK genes, the number of exons, the length of each transcript isoform, and any other key features. Moreover, these important results have to be described and commented.

Response: Following your suggestions, we have supplemented a table (Table S3) which include the number of splicing variants for each DcaCIPK genes, the number of exons, and the length of each transcript isoform. And these results were described and commented in the revised manuscript.

Comment 2: Figure 4: a hierarchical clustering was performed together with the heatmap visualization of the DcaCIPK genes expression profiles. The HClust parameters have to be mentioned (distance, linkage method). Moreover, the resulting clustering is not commented.

Response: We have added description for the HClust parameters (distance, linkage method) in the section “Materials and Methods”.

Comment 3: Page 4, line 82 “[…] their genes have introns raged from zero to four.” I guess the authors meant “ranged”

Response: Thanks for your reminding. We have corrected the spelling mistake.

Comment 4: Page 7, line 142: the authors used the words “volatile genes”. I think that, in this context, this adjective can’t be used for qualifying “genes”

Response: Thanks for your suggestion, we have rewritten this sentence into “Ten of the significantly fluctuating expressed genes…”.

Comment 5: Pages 11-12, lines 217-218: “qRT-PCR assays also validated the predicted function of DcaCIPK12 and DcaCIPK16 under drought treatment”. While qRT-PCR can’t validate a protein/gene function, please rephrase without using “function”

Response: Thanks for your reminding, we have used “predicted expression patterns” instead of “function” in the sentence.

Comment 6: Page 14, line 320: start a new line for “4. Materials and methods”

Response: We have set a new line for the title, “4. Materials and methods”.

Comment 7: “Heml 1.0”, according to the reference No 59, the spelling is HemI (uppercase “i” not lowercase “l”)

Response: Thanks for your reminding, we have corrected the spelling mistake.